# An Approach to Changing Competence Assessment for Human Resources in Expert Networks

**Mikhail Petrov** [1,2] 

[1]   St. Petersburg Federal Research Center, Russian Academy of Sciences (SPC RAS),
     Saint Petersburg 199178, Russia; mikhail_petrov@itmo.ru
[2]   Information Technologies and Programming Faculty, ITMO University, Saint Petersburg 197101, Russia

**Abstract:** An expert network is a community of professionals in a specific field, united by an information system, in which different tasks are solved. One of the main tasks in expert networks is the selection of specialists with specified competencies for joint problem solving. The main characteristic of an expert network member is a set of competencies, which includes both functional aspects and personal qualities. For this reason, the procedure for selecting specialists and ranking them is critical. Such a procedure uses specialists' competence assessments from the expert network. If these assessments are out of date, the project results can be unsuccessful. This article proposes an approach aimed at automating the assessment of the specialists' competencies based on the projects results. This approach consists of a reference model and an algorithm of competence assessment change for human resources. The paper also includes an algorithm evaluation on generated data.

**Keywords:** competence management; human resource management; changing competence assessment; expert networks; project management

---

## 1. Introduction

Competences are often defined as knowledge, expertise, skills and abilities that people need to carry out job roles. An efficient management of human and non-human resource competences prevents imposition of exorbitant costs, improves the quality of products and services and facilitates better workforce planning [1]. Expert systems solve many of the problems that arise in competency management processes. They support strategic decision making and optimize competence management by identifying unused competencies and predicting competencies. The use of expert systems provides better opportunities for career growth and increases the productivity of specialists.

Competence management is also important for project management. Project teams are increasingly assembled for temporary, specific tasks. However, such teams have to be carefully designed in order to provide successful problem solutions. Team competences are important for project teams' success. Project teams assembled to solve a task require relevant competences to deliver high performance outcomes [2].

Team members' competencies are important project success factors [3]. Experience, skills, knowledge and competence of the project manager, contractor and project group are critical to the project success, especially for the projects with considerable levels of uncertainty regarding the scope of the work, impact, methods or the outcome [4].

Efficient human resource management needs accurate assessment and representation of available competences. Any competence management system should include competence assessment [1]. However, this is a time-consuming task, so it is performed once for each specialist. Human resource managers question specialists and define their competencies based on employee's answers and certificates. In other cases, employees choose their own competencies from a predefined list. Thus,

specialists' competence profiles in expert networks do not represent competence change. This can lead to inefficient project management, which affect the success of the projects. Analysis of the project results can help to identify competence change.

This paper proposes an approach to change competence assessment for human resources in expert networks. This approach uses information about fulfillment of different project factors to identify project success. Then, it changes assessment for project participants' competencies in the expert network according to the project success. The competence change of the project participants during the project performing is not taken into account in this paper, since there are no formal indicators that can predict such a change.

The main research questions to be addressed are the following:

1. What relations between the project-performing results and competencies of the participants are needed to evaluate participants' competencies?
2. How can the project-performing results be defined in order to use them?
3. Who defines the project-performing results?
4. How can the competence assessment changes based on the project-performing results be suggested?

The approach consists of the reference model of human resource competence assessment and change, as well as the competence assessment-changing algorithm. This paper is an extension of work on the competency management systems [5–8] and method of expert group formation for task performance [9,10]. This and previous papers imply that competence management within expert networks indicates the specialists' competence assessment and its change caused by human resource managers.

The rest of the paper is organized as follows: Section 2 considers the related works in the area of human resource management, project success factors and the impact of competencies on project implementation. The third section is devoted to proposed reference model of human resource competence assessment and change. The competence assessment-changing algorithm is represented in detail in Section 4. Section 5 describes the algorithm evaluation. The conclusion summarizes the paper.

## 2. Related Work

Bohlouli et al. [1] describe competence analysis and assessment in human resource information systems. They use software technologies in conjunction with mathematical and statistical methods to assess and analyze competencies in human resource management information systems. The authors propose a flexible framework that allows experts to perform various tasks. This framework increases the efficiency of assigning specialists to projects and simplifies the hiring process.

Klarner and colleagues [2] study how competences of the team members and processes within the team affect its performance. The purpose of this study is to identify the difference in effects of the competences required by the task and behavioral competences of the team. It appears that task competences directly affect team performance, and team competences are crucial for team processes. Thus, both task and team competences are important for the project teams' success.

Oh and Choi in [3] explore interrelations of different types of team competences and how they correlate with project success. They also define project success factors and associate each competence type with them. The results of this work show the direct impact of team members' competencies on the project success. In contrast to the competencies, the team members' roles do not affect project success factors.

A framework developed by Hussein [4] distinguishes different types of projects based on their characteristics. For each of these types, it determines project success factors. The author used insights from real-life project cases to develop such a framework and to define success factors, which depend on project characteristics and context. The results described in the paper also show factors common to all types of projects.

The study in paper [11] explores the impact of a project leader's competencies on the project success at the stage of defining the project requirements. The purpose of da Silva et al. in this paper is to identify the conditions that allow the project leader to carry out all the activities.

Success factors and criteria are also identified in paper of Cserháti and Szabó [12]. Success factors are defined by them as task-oriented and relationship-oriented management areas. Success criteria contain objective-oriented and stakeholder-oriented attributes. The paper describes the relationships between them and how to use them to achieve project success.

Loufrani-Fedida and Saglietto [13] reviewed management mechanisms for competencies and projects. Such mechanisms allow one to assess and improve the competencies of individuals, groups and organizations in order to improve projects performance. The authors take into account different types of project management competences and detail fourteen micro-practices underlying them.

Omar and Fayek present an approach [14] that allows them to evaluate and model project competencies and project performance. The approach contains a methodology of project team members' competency evaluation and a model of competencies and success of the project correlation.

The paper by Hussein et al. [15] explores the links between the problems which can appear at different stages of a project. At the initial stage of the project, the problems are related to the definition of the project success criteria. At the final stage of the project, the problems are related to the assessment of previously defined criteria and the closure of the project. Thus, the correct definition of the criteria is one of the key factors to avoid problems.

Research conducted by Clevelands [16] covers successful and unsuccessful projects in various fields. The aim of the study was to identify the leadership qualities that ensure the success of projects. The authors describe skills corresponding to four leadership styles that increase the likelihood of successful project completion. The authors also propose a framework that links competencies and leadership styles.

Bachtadze et al. [17] present the competence modelling method and tools for project knowledge management. The group competence expansion algorithm is proposed based on this method. The authors of the model pay special attention to the cost of assigning competent specialists to existing teams when working on a project.

Zaouga et al. [18] describe an approach that allows one to build a common view within human resource management. The use of this approach allows for the organization of effective interaction of human resources and increases the efficiency of their use of tools and techniques. The approach implies using a knowledge base, linking roles and sets of competencies and linking processes with roles, the necessary tools and techniques and the needed artifacts.

López et al. [19] analyze existing methods for assessing the competence of human resources and propose two new algorithms for this process. These algorithms process information about projects assigned to specialists in different ways to obtain a set of their competencies based on the requirements of these projects. However, the proposed algorithms do not take into account the results of the analyzed projects.

Nikitinsky in [20] explores the possibility of improving talent management and human resource management by using data mining. The paper also describes a mechanism that analyzes an employee's documents to extract his competencies.

The analysis results of the related works considered above are presented in Table 1. The main purpose of the analysis was to identify common principles used in the approach described.

Related work analysis shows that the competencies of the project participants directly affect the success of its implementation. Some works focus on the fact that competencies related to different types affect the result in different ways. In addition, it is important to choose the right project's success criteria, which may differ for different types of projects. Cooperation of project participants and their joint competencies also plays a significant role.

Thus, the relations between the project-performing results and competencies of the participants are well studied. However, a method that takes these relations into account to change the performers'

competence assessment has not yet been developed. As the related work research showed, competence assessment in expert networks is either not updated or conducted manually, according to the subjective assessment of human resource managers.

**Table 1.** Related work analysis.

| Principle | [1] | [2] | [3] | [4] | [11] | [12] | [13] | [14] | [15] | [16] | [17] | [18] | [19] | [20] |
|---|---|---|---|---|---|---|---|---|---|---|---|---|---|---|
| Project team forming | ✓ | | | | | | | | | | | ✓ | | |
| Competence management | ✓ | | | | | | ✓ | | | | ✓ | | ✓ | ✓ |
| Team competences | ✓ | ✓ | | ✓ | | | ✓ | | | | ✓ | ✓ | ✓ | |
| Competences as a project success factor | | ✓ | ✓ | ✓ | ✓ | | ✓ | ✓ | | ✓ | | | | |
| Different impact | | ✓ | ✓ | ✓ | ✓ | | | ✓ | | ✓ | | | | |
| Different project types | | | ✓ | ✓ | | ✓ | | | | | | | | |
| Different competence types | | ✓ | ✓ | ✓ | ✓ | | | | | ✓ | | ✓ | | |
| Project success factors | | | ✓ | ✓ | | ✓ | | | ✓ | ✓ | | | | |
| Key competencies | | | | | | ✓ | ✓ | ✓ | ✓ | ✓ | ✓ | | | |
| Project KPI | | | | | | ✓ | | ✓ | ✓ | | | | | |

The approach described in this paper assumes the automation of the competence assessment change based on the principles described above. The reference model of competence assessment and change was developed for this purpose.

## 3. Reference Model of Human Resource Competence Assessment and Change

The proposed reference model of human resource competence assessment and change in the expert network is shown in Figure 1.

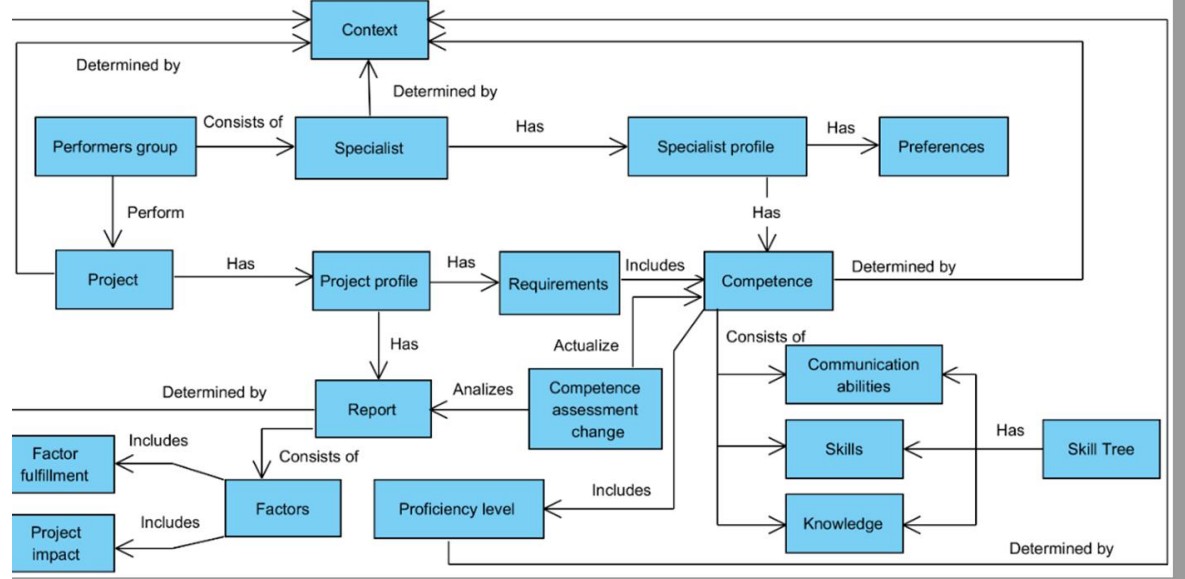

**Figure 1.** Reference model of human resource competence assessment and change.

The specialist is one of the key elements of this model. Specialist profiles contain their preferences and competence in the expert network. Specialist preferences comprise the specific specialists' peculiar properties of the work, such as the work schedule. Project managers consider them when assigning specialists to projects. Competence is a set of competencies, each of which represents the qualification of a specialist in a particular skill. Each competency presupposes possession of a certain professional skill at a certain level. Competencies are defined for each specialist and entered into the expert network by human resource managers. Professional skills can represent practical skills, knowledge or

communication abilities in a professional field. The proficiency level is determined for each professional skill separately.

　　Projects are assigned by the company's management in order to fulfill its strategic goals. Project managers operate the projects and assign performer groups for them. Performer groups consist of one or several specialists who can perform the project. The project requirements are determined for each project by the project manager. They define the competencies of participants necessary for the successful implementation of the project. The project requirements from the project profile and the specialist competencies from the specialists' profiles are matched to determine project participants. A separate group is assigned to each project so that its requirements are covered by the competencies of the specialists included in the group. The formation of a performer group to complete the task is described in more detail in [10].

　　The project reports represent the project results and consist of different factors, such as alignment with the project budget, satisfaction of stakeholders and alignment with the schedule. For each factor, the report indicates the degree of its impact on the project success (as a percentage) and the degree of this factor fulfillment. If there are additional factors influencing the assessment of the success of the project, they can also be indicated in the report. The sum of all the factor degrees of impact for each project should not exceed 100%. A combination of factors is used to determine the success of the project. The success of the project determines the increase or decrease in its participants' competence assessment. A more detailed description of the procedure for changing the assessment is described below.

　　Context determines additional information about specialists, projects, reports, competencies and its proficiency levels, such as informal description, contact information or certificates. Expert network users use the context for better decision making and cooperation. It also can be used for the assessment of project results.

## 4. Competence Assessment Change for Human Resources

　　The competence assessment-changing algorithm is schematically shown in Figure 2.

　　The algorithm consists of two stages. The first stage is the project success assessment. Each completed project must be assessed for its success. To do this, the project success coefficient is calculated based on the factors specified in the project report. This coefficient is compared with the acceptable value determined by the project manager. If the coefficient is below the acceptable value, then the project is unsuccessful; otherwise, it is successful.

　　The values indicated for each factor in the report are used to calculate the project success coefficient:

$$S = \sum_{1}^{f} \frac{I(F_i)}{100\%} \times \frac{E(F_i)}{100\%} \times 100\%, \tag{1}$$

where $S$ is the project success coefficient; f is the number of factors defined for this project; $F_i$ is the $i$-th factor in the project report; $I(F_i)$ is the degree of the project impact of factor $F_i$; and $E(F_i)$ is the degree of factor $F_i$ fulfillment. Thus, the project success assessment considers each factor's importance and fulfillment.

　　The second stage is the change in competence assessment. A change in competence assessment is proposed for each competency of each project participant.

　　First, the degree of influence of this competency on the project result is determined. It depends on two indicators (see (2)): the required proficiency level relative to other requirements, and the expert's proficiency level relative to other participants.

$$D = ((p/P) + (r/R)) / 2, \tag{2}$$

where D is the degree of influence of the competency on the project result; p is the proficiency level of the competency; P is the sum of all participants' proficiency levels in this competency; r is the

required proficiency level for this competency; and R is the sum of all required proficiency levels for all competencies in the project. Thus, the degree of influence of the competency considers uniqueness of competency among other project participants, and its importance for project performing.

The change in a performer's competence assessment depends on d and on whether the project is successful:

$$C' = \min\{C + M \times d \times s, M\}, \tag{3}$$

where C′ is competency's proficiency level after assessment change; C is competency's proficiency level before assessment change; M is maximal competency's proficiency level; s is 1 if the project is successful; otherwise, it is −1. If C′ is more than 0 and less than 1, then C′ = 1. If C′ is less than 0, then the competence is removed. Thus, competence assessment decreases or increases in relativity to the maximum level of the competency, taking into account the degree of influence of the competency on the project.

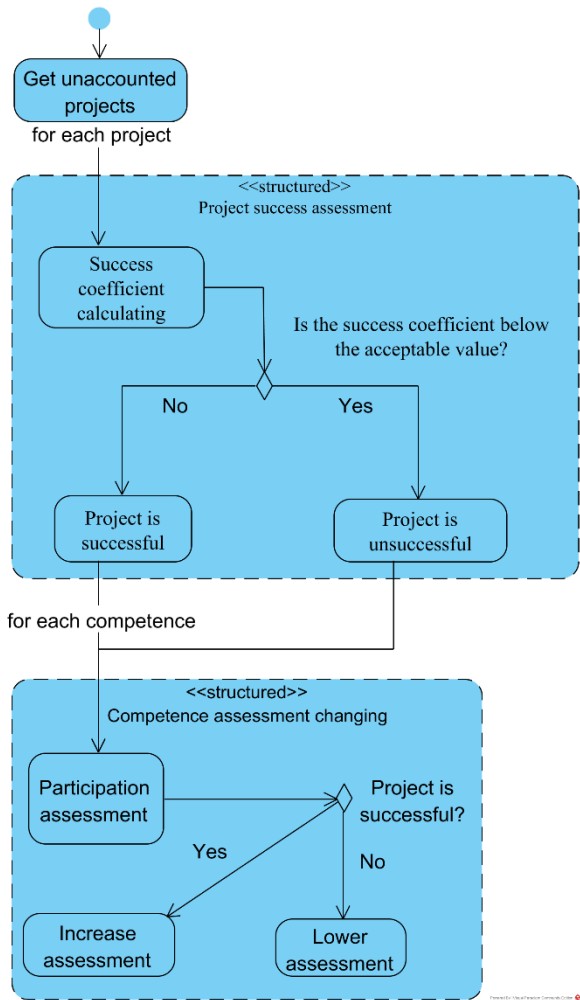

**Figure 2.** The scheme of the competence assessment-changing algorithm.

## 5. Algorithm Evaluation

The competence assessment-changing algorithm evaluation is aimed to discover and explain patterns in the results of the algorithm work for different sets of data. The generated data were used to do this. The data set generation details are described in Section 5.1. The evaluation results and its analysis are given in Section 5.2.

### 5.1. The Evaluation Method

The data sets for the competence assessment-changing algorithm evaluation should contain projects with different numbers of requirements and participants. Such data sets are able to show the dependence of the algorithm results on these parameters.

The generated projects should look like real ones and be performable. The generation parameters were chosen taking these conditions into account. Each project contained from 5 to 10 requirements and from 2 to 7 performers. The performers' group formation algorithm described in [10] was used to assign performers to the projects. This algorithm assigns specialists on the project according to their competencies and the project requirements. It chooses only those specialists who are available and have at least one competence that is needed and that no one on the project group has. Thus, each project team had a set of competencies required to perform the project. The proficiency level for all competencies was set from 1 to 6 based on the paper [8]. Such division covers different levels of proficiency, from basic proficiency to deep specialization. The required proficiency level for the generated requirements was set from 1 to 3, because too high requirements would complicate the performers' group formation.

The generated data sets contained 15 projects. The projects' success was determined randomly, so 6 "successful" and 9 "unsuccessful" projects were created. The average number of requirements was 7.53. The average number of project participants was 5.4. The average proficiency level among competencies was 4.4, and the average proficiency level among the requirements was 1.94.

### 5.2. The Evaluation Results

The competence assessment-changing algorithm was applied to the generated data sets. As a result, project participants' competence assessments were changed, on overage, by 1.08. Average competence assessment changes among the projects with a different number of participants and among the projects with a different number of requirements are shown in Figures 3 and 4. Full algorithm evaluation results are shown in Table 2.

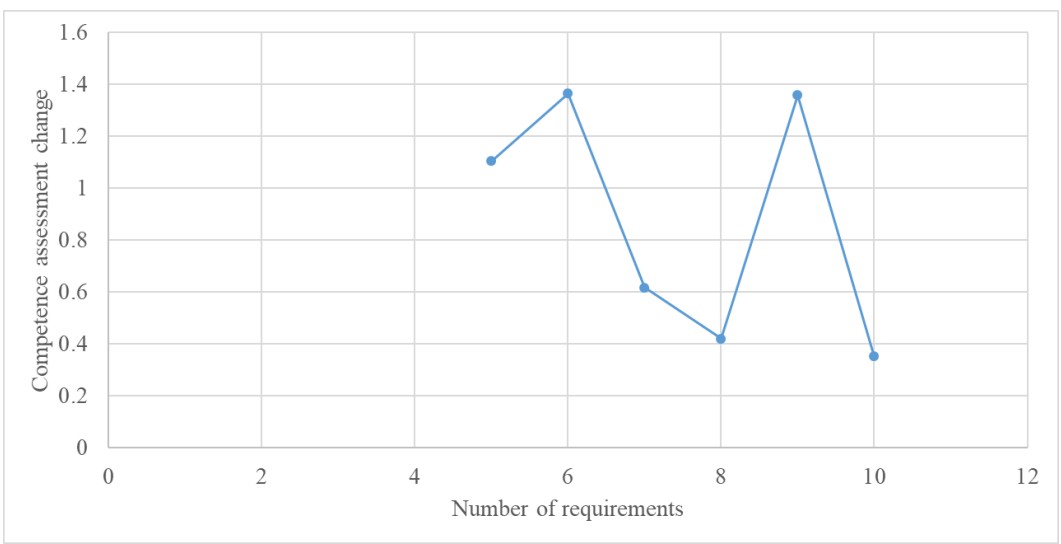

**Figure 3.** Average competence assessment changes among the projects with a different number of requirements.

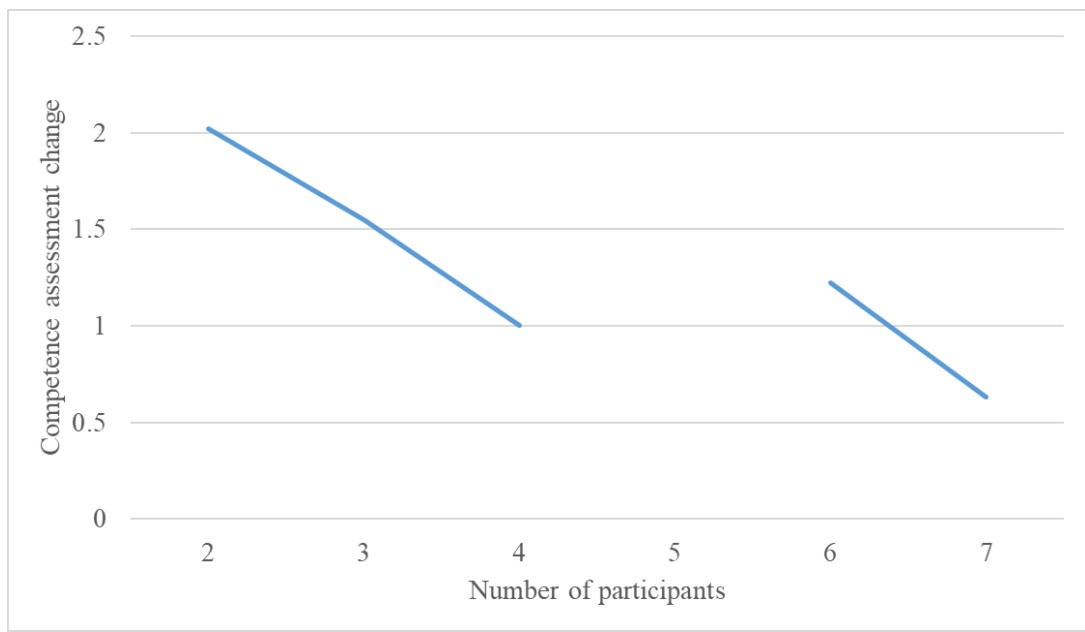

**Figure 4.** Average competence assessment changes among the projects with a different number of participants.

**Table 2.** Algorithm evaluation results.

| Project | Number of Requirements | Number of Participants | Average Competence Assessment Changes |
|---|---|---|---|
| 1 | 10 | 7 | 0.352941 |
| 2 | 5 | 6 | 1.230769 |
| 3 | 9 | 6 | 1.2 |
| 4 | 7 | 4 | 1 |
| 5 | 5 | 7 | 0.583333 |
| 6 | 6 | 6 | 1.230769 |
| 7 | 8 | 7 | 0.421053 |
| 8 | 9 | 2 | 1.6 |
| 9 | 9 | 7 | 0.95 |
| 10 | 5 | 3 | 1.5 |
| 11 | 9 | 7 | 0.363636 |
| 12 | 9 | 3 | 1.6 |
| 13 | 6 | 7 | 1.5 |
| 14 | 7 | 7 | 0.235294 |
| 15 | 9 | 2 | 2.444444 |

The horizontal axis in Figure 3 shows the number of requirements. The horizontal axis in Figure 4 shows the number of participants. In both charts, the vertical axis shows the average competence change.

*5.3. Evaluation Discussion*

As the charts show, the competence changes do not depend on the number of project requirements and are inversely proportional to the number of participants. This demonstrates the distribution of responsibility among the participants. However, this may be due to the difference in the average proficiency level among competencies and requirements. Further evaluation on the real data and research should clarify this issue.

The evaluation results also show that the biggest changes in competence assessment are made in small project teams. Consequently, real data on projects with different numbers of participants are needed for further evaluation.

## 6. Conclusions

An approach to a change in competence assessment for human resources in expert networks is developed and described in this paper. It includes the reference model and the competence assessment-changing algorithm, which are also described. The reference model allows one to keep information about different project implementation factors and their fulfillment and impact. Thus, different project success criteria for different project types are taken into account by the model. In addition, the model allows one to keep and match information about specialists' competencies and projects requirements. The competence assessment-changing algorithm analyzes the project implementation factors to identify project success. It uses this information and information about the influence degree of each project participants' each competency to change the competence assessment of the specialists in the expert network. The evaluation of the algorithm showed that the result of its implementation corresponds to the expected ones, that is, the assessments of the specialists' competencies are changing in accordance with the amount of work assigned to them.

The approach provides answers to the questions posed by the analysis of the related work. It establishes a strict formal link between the project success factors and the competencies of its participants. At the initial stage of the project, the project manager determines its success factors and indicators that allow for the calculation of the success of the factors' fulfillment. The approach uses these data to calculate changes in the project participants' competencies, which are proposed to the project manager after its completion.

Using the approach described in this article will keep the human resources' competence assessment up to date. All inconsistencies in the competence assessment that affect the results of projects success are taken into account and eliminated by this approach.

Further work involves the implementation of the proposed approach and its testing on real data. At the same time, adjustments and additions to the model and the algorithm are possible.

**Funding:** The presented results are part of the research carried out within the project funded by grant #19-37-90094 of the Russian Foundation for Basic Research. I. Algorithm evaluation has been partly supported by Russian State Research # 0073-2019-0005.

**Conflicts of Interest:** The author declares no conflict of interest.

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
