# Peer review of "An Approach to Changing Competence Assessment for Human Resources in Expert Networks"

_futureinternet, doi:10.3390/fi12100169_

Round 1

Reviewer 1 Report

Nowadays projects are the main tool to develop new things since experts that have the experience to solve together some tasks are joined together for a temporary group. Expert networks provide possibilities to manage these groups to describe experts using well-known competencies methods and define the best groups for project implementation. However, the competence-based description is expert-dependable since only he/she knows exactly about his/her competencies. The author of the paper proposes an approach to competence assessment based on project participation and its success. The author proposes a math-based approach as well as synthetic experiments for competence assessment.

The paper is interesting and well organized. However, it has a lack of related to the experiments. It would be nice to see experiments with the real people and their estimations about precision of the presented approach.

Reviewer 2 Report

The topic of this manuscript could potentially be interesting for the readers of the Future Internet journal. However, in my opinion the manuscript in its current form lacks the potential that could be achieved. I suggest that the authors make the following changes to the manuscript:

  1. The research gap is somehow addressed in the Introduction section (sections 5 and 6), but it should be better explained to the readers. Authors have listed several key concepts that are important for project management and are already known from the literature, and this is OK. However, I suggest that the authors give more explanation in the field of systems for competence assessment regarding of what is already known from the scientific literature, and how they contribute to the field.
  2. The main research questions are not addressed and are completely missing. Authors should provide and explain it in the introduction part of the paper or following the section two.
  3. In section two authors provide summary of previous papers. They should improve the style of writing, since they are starting every paragraph with the similar phrases, such as “Paper [x] identifies,… Paper[x] presents…, paper [x] propose…, etc.”
  4. It seems to me that the literature review is underdeveloped. Besides focusing on the literature addressing the concepts and competences that are evaluated inside the system, authors should, if possible, present other existent models of competence assessment and changing in the field of human resource.
  5. I am not shore if the phrase “Reference Model of Competence Assessment Changing for Human Resource” is appropriate. In my opinion, it should be named “Human Resource Reference Model of Competence Assessment and Changing”. Please consider revising it throughout the whole manuscript.
  6. Conclusion section is underdeveloped and only reviews what is already known from the manuscript. It should be improved. What possible new insights are provided by the study and how does the research results differ or are similar to previous studies? What are the possible managerial implications and limitations of the study?
  7. Style of writing should be improved and manuscript should be proofreader by the expert English editor.
  8. Manuscript should be prepared according to the instructions – see research manuscript sections in instruction for authors: Introduction, Materials and Methods, Results, Discussion, and patents (if applicable).

Reviewer 3 Report

This paper proposes an approach aimed at automating the assessment of the specialists’ competencies based on the projects results. This is achieved by employing a reference model and an algorithm of competence assessment changing for human resource. The paper also includes algorithm evaluation on generated data. The theme of the study is quite interesting and suitable for publication in the Future Internet Journal.

The structure of the paper is logical. Section 1 gradually introduces the problem addressed by the study. The literature review section surveys several previous studies that relate to the subject. I find quite strange the way the author sites previous work in this section. I would suggest changing in a more usual manner. For example in the beginning of section 2.  Bohlouli et all. [1]  describe ……. The main problem with sections 1 and 2 that they exhibit an extremely high degree of similarity with text that are available online.

In section 5 the subsections have wrong numbering (5.1 and 5.2 instead of 3.1 and 3.2).

The evaluation section is quite limited in size. I would suggest expanding it.  The evaluation method is very simplified since it based only in a set of generated data that exhibits specific trends. One question is how those data were generated? I do not believe that a simple reference to a previous work is sufficient. The reader should be able to fully understand the study without having to read additional studies.

Thus, the evaluation it is not based on actual data. I was also wondering on how easy is to acquire such data. And if it is difficult how practical is the proposed model?

 I would like to see some elaboration on the previous questions since to my opinion they constitute significant limitations of the study. One way to strengthen the argument of the generated data is to run multiple evaluation with different sets of generated that should have some specific reasoning for their actual form.

 Also, the results of the evaluation are not discussed. I would like to see a thorough discussion in order to support the value of the study.

Round 2

Reviewer 2 Report

Final version of the manuscript should be proofreaded by the expert English editor.

Reviewer 3 Report

None